# Rethinking Women's Leadership Development: Voices from the Trenches

**Robin Selzer [1],\*, Amy Howton [2] and Felicia Wallace [3]**

1   Experience-Based Learning & Career Education, University of Cincinnati, Cincinnati, OH 45220, USA
2   Design Impact, Cincinnati, OH 45202, USA; amyjhowton@gmail.com
3   Academic Excellence and Support Services, University of Cincinnati, Cincinnati, OH 45220, USA;
    Felicia.Wallace@uc.edu
*   Correspondence: Robin.Selzer@uc.edu

**Abstract:** As recent graduates of a women's-only leadership development program in higher education in the United States, we used autoethnography as a research methodology to provide critical insight into effective women's leadership programming and evaluation. The potential of this methodology as both a learning process and product helped elucidate two key findings: (1) to effectively develop women leaders, work must be done at the personal, interpersonal, and organizational levels, as these levels are interrelated and interdependent; and (2) women's multiple identities must be engaged. Therefore, relationship-building should be a central learning outcome and facilitated through program curricula, pedagogical methods, and evaluation. Including autoethnography as a program evaluation methodology fills a gap in the literature on leadership development, and supports our goal of making meaning of our personal experiences in order to enhance women's leadership development.

**Keywords:** women's leadership; leadership program evaluation; gender equity; intersectionality; identity; higher education; career advancement

---

## 1. Introduction

As recent graduates of a women's-only leadership development program in the United States, we came together to make sense of our own experience, and get to know each other better. Through the process, we recognized the need to define women's leadership development, so that we had a common understanding. Ely, Ibarra, and Kolb's [1] definition of women's leadership development resonated with us—we used this as a guiding framework for our meaning making. They reference leadership development as identity work, and offer three guiding principles for successful women's leadership development programs: (1) considering topics in light of gender bias; (2) supporting women's identity work; and (3) focusing on leadership purpose. This speaks to our own needs, related to our experiences as women's leadership development program participants. Our identities as women leaders are tied to our sense of purpose, and are uniquely informed by our gender and other dimensions of our identities. We came together to explore these topics for ourselves and to figure out how to make a contribution to other women in our shoes. As we did so, we wondered how the program design we participated in provided a sense of purpose, engaged our identities, and created a space to express those things with others at the interpersonal and organizational level.

## 2. Cracks in the Glass Ceiling and Still Feeling the Chill at University X

In the current United States climate, discussions about women's leadership have been pervasive, with thanks in part to best-selling books on the topic, such as Sheryl Sandberg's [2] *Lean In*, which have

been endorsed by famous women leaders, such as Oprah Winfrey. The topic of women's leadership has also been at the forefront of popular culture due to high profile magazines, like New York Magazine, covering stories that ask questions such as whether Marissa Mayer (CEO of Yahoo) can have it all [3]. The Atlantic [4] also published an article that went viral on social media with the same headline, entitled "Why Women Still Can't Have It All." On the one hand, women are advised to lean in [2] and on the other, lean back [5]. Now, women are being told to be disruptive [6]. Much of the literature on women's professional development focuses on the need for mentors. Yet, many women's leadership experts propose that women forget about mentoring and find a sponsor to break the glass ceiling [7,8]. Needless to say, women are receiving many mixed messages on this topic.

Despite all of the discussion on how to be a woman leader, have it all, and do it well, gender disparities in higher education remain intact. Dixon [9] asserts that a chilly climate still exists. According to Cook [10], only 26% of college Presidents in the United States are women. Outside of the United States, the equivalent leaders of a college or university are also known as Chancellors or Rectors. Fink, Lemaster, and Nelson [11] asserted that the "glass ceiling is firmly intact" (p. 59). However, some have said that women have cracked the glass ceiling. Even though we have seen an incremental rise in the number of college women Presidents, challenges based on gender bias exist. For example, Dr. Angela Franklin, the first female and African American President of Des Moines University, spoke about how women in leadership are under different scrutiny than males [12]. She speaks of women being in the double bind position [13] of damned if you do, damned if you do not, where women are either too strong or too agreeable. It is important to note that these cracks in the glass ceiling can also be viewed as even smaller splits if other demographics, such as race, class, and sexual orientation are taken into account. Only small groups of women are squeezing through, and when they do, they are still subject to very constrained ideas of what leadership looks like, and who fits that mold.

In the Guardian of Higher Education [14] article "Why Women Leave Academia and Why Universities Should be Worried," Curtis Rice points to findings from a 2011 report, "The Chemistry PhD: The Impact on Women's Retention", which speaks to gender disparity among PhD candidates wishing to continue with careers in academia. The report concludes with findings specifically related to gender: first, women are told that their gender will be a barrier to success and second, these warnings are validated by the limited representation of women in the academy, particularly in leadership roles. PhD candidates interpreted this to mean that women must make incomparable sacrifices, both personal and professional, to be an academic. The under-representation of women in senior leadership positions is not unique to the United States. For example, Louise Morley has explored this phenomenon and showed that greater gender gaps exist in the professoriate in the United Kingdom. For example, in 1994, men held 80.9% of the positions and women held 19.1%of the positions [15]. In 2013, women held still just 4415 (22.4%) of 19,745 professor positions in the UK (Higher Education Statistics Agency, 2014). In addition, Fitzgerald and Wilkinson [16] examined the absence of women in senior leadership positions across Australia and New Zealand. Similar to the United States, more women are enrolling in universities in these countries, but the gender gap exists at top levels of leadership. O'Connor and Goransson [17] examined gender in university management in Ireland and Sweden. Sweden seems to be an anomaly, with real change actually being accomplished on these issues [18]. Ireland, on the other hand, is still grappling with traditional, essentialist gender stereotypes. The heads of institutions of higher education throughout Europe are slightly below 16% [17]. Essentially, "it would seem that gender inequality in higher education is a subtle but pervasive global problem" [16]. Therefore, research into gender inequalities in higher education continues to be needed to see the full picture of inequity.

Here at our own university in the United States, these problems persist. To help assess campus climate, Status of Women Reports were conducted by the campus-based Women's Center. Ultimately, the goal of these reports was to produce recommendations focused on strategies that would, ideally, work toward creating gender equity. Data shows that the percentage of women faculty at the University

X has increased, but has yet to reach 50 percent. Women are not advancing through the ranks at the same rate as their male counterparts, particularly at the senior most ranks, such as Deans and Department Heads [19]. In comparing University of X data from 1998, 2005 and 2009, there is continued documentation of a dearth of women of color at the senior administrative and academic levels and within the faculty.

To further this benchmarking discussion, the University of X Status of Women Report [19] states that faculty women are disproportionately represented in non-tenure track positions, compared to their male counterparts. Men are also much more likely to have tenure. Approximately 63% of tenured faculty is male, while about 44% are female. These statistics parallel those nationally, which according to Catalyst.org, are at 62% and 44%, respectively. The University of X has experienced a significant decline in tenure track positions, reflecting the national trend in higher education to utilize adjunct instructors. This trend is gendered insofar as most instructors hired as contingent faculty are women [20]. Assessing the status of women staff was noted as challenging in this report, due to the lack of a centralized reporting system. Disaggregating the data according to race and gender is also challenging for similar reasons.

Beyond the creation of these reports, the University of X has addressed pervasive gender disparities in several ways. In 2005, under the leadership of its first woman president, a commitment was made to allocate resources and secure staff time to administer the Women's Initiative Network (WIN). WIN was formed in 2001 to advance gender equity at the University of X and ran a Women's Institute for Leadership Development until 2014. It is currently run out of the Provost's office. For the purpose of addressing ethical implications, the pseudonym WLP will be used as the name of this program throughout this paper. WLP was first offered in 2000, and has undergone program assessment that moved it from a multi-institutional consortium to an internal leadership program, focusing on the original program's intent to assist University of X women in advancing to senior academic and administrative positions. With new and focused institutional attention on succession planning, bringing WLP home made sense.

There is still more work to be done at the University of X and beyond. Action is needed to keep the conversation moving forward and to work towards gender equity. As Rice [14] aptly posited, "The answers here lie in leadership and in changing our current culture to build a new one for new challenges. The job is significant and it will require cutting edge, high-risk leadership teamwork to succeed" (para. 15).

## 3. Personal and Structural Barriers to Women's Leadership

Barriers to women's leadership fall into two categories: personal and structural. Personal barriers are related to self-efficacy, or one's beliefs about what she is capable of [21]. Vital to leadership development, Bandura [21] argues that efficacy is required as a foundation for all other methods of development to function or thrive. These types of barriers are addressed in more contemporary women's leadership literature like Sandberg's [2] *Lean In*. Another commonly cited personal barrier is "the conflicting responsibilities of home and family" [22]. This personal barrier is intricately connected to structural barriers. Ways to overcome structural barriers are policy-oriented, such as pausing the tenure clock or providing flexible work schedules. While barriers for women have become "more permeable" [23], structural barriers that are discriminatory still impede the advancement of women's career trajectories. In short, you can teach women about negotiation skills in a women's leadership development program, but if exclusionary policies and practices remain in place, concrete walls will continue to exist [23].

There are some proven support strategies for addressing barriers to women's leadership and success. For example, "mentoring has been cited as a long-term practice that typically involves a more senior-ranking professional providing guidance and support for a less-seasoned professional" [24]. However, mentors may not be enough. According to Schulte [8], the practice of sponsorship also provides a possible solution. While a mentor provides advice and coaching, a sponsor is someone

with a higher authority and influence in an organization that is willing to advocate and promote the upward mobility of an individual [25]. Unfortunately, women are not sponsored enough, perhaps because they may view "networking as inauthentic and akin to using people" [1].

In addition to these specific strategies, other questions have emerged in the literature on leadership development, such as the value of women's-only programming [26], how to engage women's full and multiple identities [27], the role that women's leadership development programming plays in creating organizational and institutional change [28], and the impact of program evaluation [29]. These debates were central in our autoethnographic analysis. Therefore, they will be explored in-depth later in the Discussion and Recommendations sections.

## 4. Theoretical Lens: Connected Knowing

Following our WLP experiences, the three of us came together in the spirit of reflective discourse because we wanted to learn from our shared experiences in this women's-only leadership development program, and to get to know each other better. We sought to make sense of our own experience and make meaning of what a program like this meant to us, personally and to our institution. As we collected data through our personal storytelling, we wanted to allow the project to be as organic and meaningful as possible—goals that became increasingly important to us throughout the process as we strengthened relationships with each other and drew personal value out of this research process. To guide our data analysis, we drew from Belenky, Clinchy, Goldberger and Tarule's theory of Connected Knowing, which centers the notion of the value of women listening to their inner voice. In their ground-breaking book, ***Women's Ways of Knowing*** (1997), Belenky et al. [30] found that "women repeatedly used the metaphor of voice to depict their intellectual and ethical development; and that the development of a sense of voice, mind, and self were intricately intertwined" (p. 18). They also use the concept of voice to challenge current psychological theories of human (i.e., white, male) development and to include women's stories and experiences [31]. Connected Knowing is about finding patterns in women's voices, which was a goal of our inquiry.

Moreover, Connected Knowing gives value to women's contributions as emotive, intuitive, and personal. Belenky and Stanton's [32] words clearly resonated with us when thinking about identifying a theoretical foundation, "Women have had to struggle to make their own viewpoint heard, even to themselves." (p. 78). This approach also values that learning is relational, and explores how women gain their voices and construct knowledge. However, we also recognized that Connecting Knowing felt limiting in its essentialist assumptions about gender norms and as a result, arguably perpetuates these norms. Nevertheless, as we struggled to make meaning and evaluate our experience, we found the affirming framework to be a helpful tool of analysis in its centering and in valuing our voices. The framing process for our project was inherently collaborative, driven by deep empathy. "We call them Connected Knowers because they actually try to enter into the other person's perspective, adopting their frame of mind, trying to see the world through their eyes." [32]. Our initial attempts to evaluate through examining program survey data did not suffice because it felt superficial, not getting at real issues of identity shifts and changes in beliefs. We needed time to find our voice and process our experience in a deeper way—through Connected Knowing, our truths emerged through care and empathy. Care is the quest for understanding and you have to listen for it. The most trustworthy knowledge construction comes from personal experience. Therefore, in concert with Mezirow's [33] theory of transformative learning, we chose to reflect critically with each other to interpret our experiences and shift the focus to deeper learning for continued personal growth and development.

## 5. Methodology

It is important to begin by describing the WLP program in detail. As an urban, public, top-tier research institution with multiple campuses, the University of X espouses strategic efforts to develop women's leadership, such as the WLP program. The program is part of a larger system-wide change effort to support gender equity on campus. During our participation, WLP was a seven-month

leadership program that aimed to prepare mid-career faculty and staff women for senior leadership in higher education, and to identity participants' readiness for higher-level positions within University of X. According to Hawthorne Calizo [24], it is one of the only campuses in the country that offers joint women's leadership development programs for faculty and staff. To this end, the WLP program curriculum includes self-assessments and research-based content focused on critical skill development. The goals of the program included increased self-awareness and competencies in the following broad leadership development components: visioning and strategic alignment, finance and operations, understanding and building culture, and negotiation, teams, and conflict.

The program is exclusive and the selection process competitive. Criteria include: completion of a Master's degree, the position of a certain rank- (faculty applicants must have associate, field service, or clinical status); staff applicants must be an Assistant Director or above, earning a certain pay grade, and three years of completed employment with the university. Since the program's inception, participation has been open to both faculty and staff thereby fostering "a new dialogue and a cross-fertilization of ideas" [11]. The application requires submission of a resume or curriculum vitae, a letter of recommendation from a supervisor, and a statement of interest that outlines how the program can support the applicant's development. If accepted into the program, applicants' home departments invest $100 of financial support.

The WLP Steering Committee, comprised of women senior leaders and WLP alumnae from across the university, reviews the applications and selects candidates. WIN publicizes the class of WLP participants each year through university news, various listservs, and on the WLP website. The model is cohort-based, and 10–20 women on average participate each year. Participants are enrolled in a Blackboard group to organize communication. We were all thrilled to be selected as participants, among the competitive pool of applicants for our respective years' cohorts.

During the time we participated in WLP (2011–2012 and 2012–2013), the program met for 3.5 h monthly and featured both a didactic speaker format and interactive methods, such as case studies and group networking. During the first meeting of 2012–2013, as group guidelines, participants agreed upon community expectations to engage, demonstrate authenticity, provide feedback, honor confidentiality, and be collegial. During that same cohort year, Strengths finder was the chosen self-assessment tool. We were asked to complete this self-assessment before the program began, and results were explored in small groups during the first meeting, but not discussed further. Each WLP session provided us the opportunity to explore topics on leadership strengths and style, authentic leadership, conflict management, budgeting practices in higher education, negotiation, and navigating the culture at the University of X. Speakers included senior level leadership at the university, such as the Provost, Vice President of Finance, Associate Provost for Diversity and Inclusion, and external consultants. Assigned readings were limited and used to supplement speakers' presentations on topics such as authentic leadership, crafting a vision, and performance-based budgeting. The sessions highlighted the specific strategies and behaviors that are critical in effective leadership. For instance, during the session on Authentic Leadership, we were able to work through exercises that helped us identify our existing and potential support networks. We looked at significant events in our past, how these events helped shape our priorities, and potential future events. Program sessions sought to promote the mindset and competencies necessary to transform ourselves from effective colleagues and bosses to successful and valuable leaders. Articulated learning outcomes for the overall program included building critical networks and partnerships, deepening members' knowledge of strategy, negotiation, communication and leadership, and maximizing influence with internal and external departmental stakeholders at the university.

A unique feature and pedagogical tool the WLP program offered in 2012–2013 was the experiential learning component, known as the "stretch assignment." The two of us that participated in this cohort collaborated with each other, organizers, and internal stakeholders to find a project that would help stretch us beyond the scope of our current professional role. The overarching goal of this assignment was to pull together the newly emerging theories and lessons about leadership from this programmatic

experience and apply them in relevant and meaningful ways. Each participant's project differed by genre and academic commitments according to our personal need, but connected to professional goals, hoping to increase visibility at our university. At the end of the seven months, we were offered the opportunity to share our stretch project in an informal speed-dating type format with WLP alumnae to obtain feedback. Abstracts of stretch projects were shared with all participants. Upon completion of WLP, all participants are provided with a certificate, and an email is sent to supervisors in acknowledgement of completion. In terms of program evaluation, the WLP program collects hand-written, Likert scale and brief open-ended evaluative feedback on each session from participants, and continues to make improvements based on the findings for the following year.

## 6. Autoethnography

Because our research centered on our personal narratives, we employed autoethnography as our research methodology. As mentioned, we came together to make meaning of our experiences, with a desire to contribute to making the program better for other women. We were interested in exploring our individual experiences in context with others, and autoethnography facilitates that exploration. Autoethnography allows for a wider and deeper understanding of what constitutes quality women's leadership development programming. It also aligns with the theoretical lens of Connected Knowing. Ellis and Bochner [34] define autoethnography as "an autobiographical genre of writing and research that displays multiple layers of consciousness, connecting the personal to the cultural" (p. 739). At the time we were collecting data, we did not know about Collaborative Autoethnography (CAE). CAE is actually a more accurate description of process of social inquiry because it is "engaging in the study of self, collectively" [35]. Furthermore, in CAE, the group explores the social and cultural meaning of their experiences. For the purposes of this paper, we refer to autoethnography as our methodology, with the understanding that CAE is a legitimate method that could be recommended to explore the phenomenon of women's leadership development programs as well.

We initially operationalized the autoethnographic methodology by developing four guiding questions to frame our narratives. Then, we each created written responses.

(1)    Why did I apply to the program?
(2)    What were my expectations of the program?
(3)    How did I experience the group dynamics of being in a women-only space?
(4)    What was the impact?

After writing responses to the questions, we came back together as a group to analyze our findings, processing our common and disparate experiences. As Dreschler Sharp, Riera, and Jones [36] note, the process of autoethnography helped us to "seek and find meaning in the stories that was deeper and more substantial than what we would find on our own" (p. 330). In contrast to quantitative research, the purpose of autoethnography is not to collect large samples, generalize the findings, nor compare programs. It is not a finite answer to a single question, but an investigation of a lived experience. Therefore, it is appropriate that this study be locally focused, using one location to deeply examine a set of experiences. It is part of the research design.

At the same time, we understand that our research questions and findings are not in isolation. We know the personal is political. We understood this methodology as feminist, helping to make connections between the personal and the structural. We worked to place our lived experiences at the center of the research process, thereby moving our voices from margin to center. Importantly, this was a challenging process and we struggled along the way to find and share our voices, even in terms of writing. For example, we kept shifting from first to third person throughout initial drafts. Our struggle with voice speaks to the power and potential in this methodology to center voice, both theoretically and practically. Our conversations and corresponding analysis became more than our personal narratives—they began to shed light on issues of larger institutional and structural gender inequity.

True to autoethnography and our theoretical framework, our personal identities informed the entire research process. As co-researchers, we share several key, social identities: age (all in our forties); motherhood (we all have children); married to men (one now divorced); holders of doctoral degrees; of similar socio-economic class (given that we are working in the same institution in comparable positions, with comparable levels of education); and geographical location. One key dimension of difference is race: one of us is African-American; two of us are White. We are also all staff, though from different areas of the institution including academic advising, student life, and academic student support services.

*Ethical Considerations*

In terms of the ethical approval for the study, autoethnography as a method has interesting considerations because one is writing about self and not necessarily studying others. IRB approval isn't necessary in all cases for this very reason. However, Tolich [37] raises important points about the "rights of others weighted against the interest of the self" (p. 1599). For example, our stories have the potential to identify staff involved in the design and delivery of the WLP program, as well as the University of X. We contemplated this early on, and considered whether we would do harm to our self or others. Ultimately, we concluded that ethical issues did not exist because risk was minimized. For instance, in terms of harm to self, none of us used the autoethnographic process to heal ourselves. One author has since left her professional position, and the field of higher education as well. In terms of harm to others, we agreed that risk was minimized because several years have passed since our participation and the WLP program has changed in structure and in leadership. The program has a new name and Director. In fact, an early draft of this paper was shared with the new Director for full transparency.

As Boyle and Parry [38] remind us, the autoethnographic process is intensely reflexive in nature. We found this to be true, and that analyzing our WLP experience in this way was necessary for us to make sense of it. Our focus was also to dialogue about improving the culture of the program. The point of autoethnography is to move from the local personal analysis to the larger organizational analysis. This felt inherently feminist, whereby the personal is seen as the political, where the status quo is challenged, and marginalized voices move to the center. Still, it is important for all involved to reflect on the role of the writing process so that one voice isn't privileged over others and co-generative dialogue exists [39]. In this study, we do not use pseudonyms because we have not identified which author's words are quoted. This protects anonymity.

To be sure, there are debates about ethics and autoethnography [40]. We understand that once the study is published, we may lose this anonymity and also connect ourselves to the University of X. We understand that by sharing our voices we become vulnerable; and this poses a professional risk due to our stories evoking emotions. Yet, we see this vulnerability as strength, and know we speak only for ourselves as participants intimately engaged in the experience. Just as we have provided a critique and analysis, we know we must allow the same for our stories. These stories are important to tell to help women and women's leadership development programs. In fact, WLP has made changes in the program structure since our participation, which eliminates some of this risk.

Throughout the writing process, we discovered that we need others to help demarcate our own personal and professional paths to success. Without the WLP program experience, the three of us would not have come together in this research project, with the shared goal of helping to rethink women's leadership program development and evaluation. This process allowed us to create a safe space to both acknowledge vulnerabilities and struggle with the idea of how to have it all. It was an intellectual and emotional process. The ability to take risks to share our authentic stories while writing with other supportive women was therapeutic and instrumental in shaping how we viewed ourselves as women leaders. We created a writing space and process that helped to hold us accountable to each other from a place of caring where flexibility was valued. As we continued to meet, these guidelines were renegotiated along the way. For transparency purposes, we discussed the need to be

fully committed, accountable, timely, responsive, communicative, direct, not take things personally, and not deterred by personal conversations or social media/email/text distractions during work sessions. Rather than informed consent, we practiced what Tolich [37] calls "process consent", where consent to continue participating was voluntary and negotiated along the way. Even as this paper was submitted for publication and undergone revisions, we made sure each of us still wanted to participate.

Exploring how to write together on this piece helped to develop insights on collaboration and collective productivity. For example, we found a way to honor each other's processing style. We began writing over a year ago and the prioritization of producing the piece was difficult to justify with so many competing demands. In our writing group, we found a space in which inclusion and unique ambitions for success were met with support as we strived to meet our personal and collective goals. Hearing each other's stories and being reminded of our goals and successes to date provided needed strength and support.

In terms of analysis, we discussed similarities and contradictions in our personal narratives and one of us served as a recorder to transcribe our thoughts. As we shared our stories, we used Connecting Knowing to test our assumptions and further explore each other's experiences, which led to a deeper understanding. The metaphor of voice ended up depicting our ethical development, just as Belenky et al. [30] had stated. We acknowledged that our racial differences might impact our different experiences. This sparked conversation not about the strengths of the WLP program, but what was missing but expected to be addressed. In the following section, we interrogate this idea further and key themes are addressed.

## 7. Results

Engaging in autoethnographic methodology led to the understanding that in order to effectively develop women leaders, work must be done at the personal, interpersonal, and organizational levels; and that these levels are all interrelated and interdependent. Within these areas of development, eight emerging themes rose to the surface. At the personal level, (1) vulnerability, (2) structured space for reflection, and (3) being seen and heard; at the interpersonal level, (4) intersectionality, (5) supportive community for vulnerability and (6) networking; and at the institutional level, (7) cultivation of authentic leadership and (8) changed culture (See Figure 1). In sum, women's leadership development starts with personal reflection, involves engaging with other in identity work, and needs structural and institutional support.

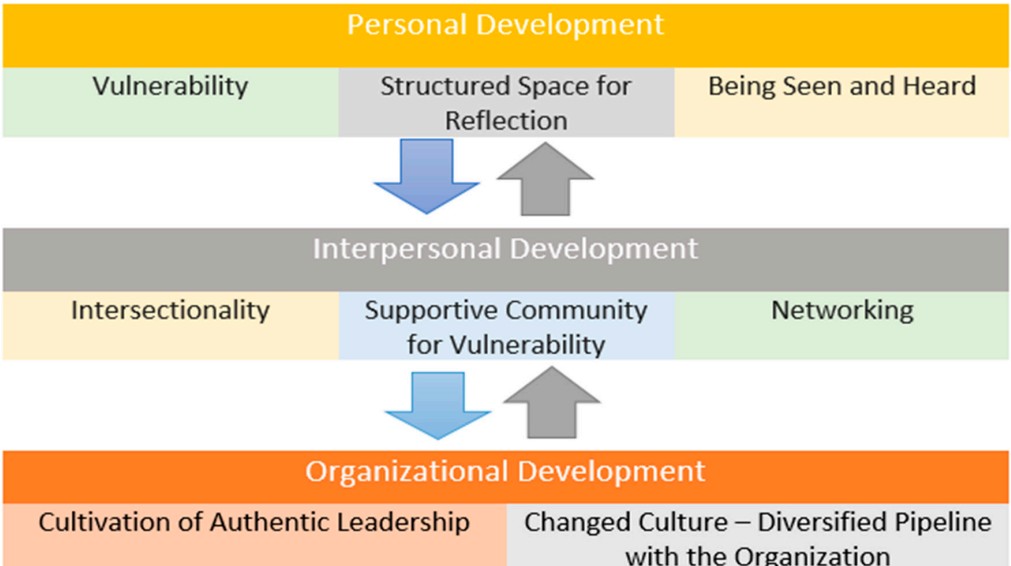

**Figure 1.** Women's Leadership Levels and Themes (Final Version).

Initially, our findings at these three levels were indicated by the themes in Figure 2. At the personal level, credentialing, visibility, career assessment and empowerment emerged as themes. At the interpersonal level, networking, shared experience, and support were significant. At the organizational level, cultivation for leadership and being a part of the professional pipeline were central. At that early point in our analysis, we agreed that our professional development was positively impacted by our participation in the WLP program, and that the program was successful in its goal of creating a leadership development opportunity for women at the institution. However, looking back, at this initial stage of our research process, we were in a very guarded, superficial space—both with ourselves and each other.

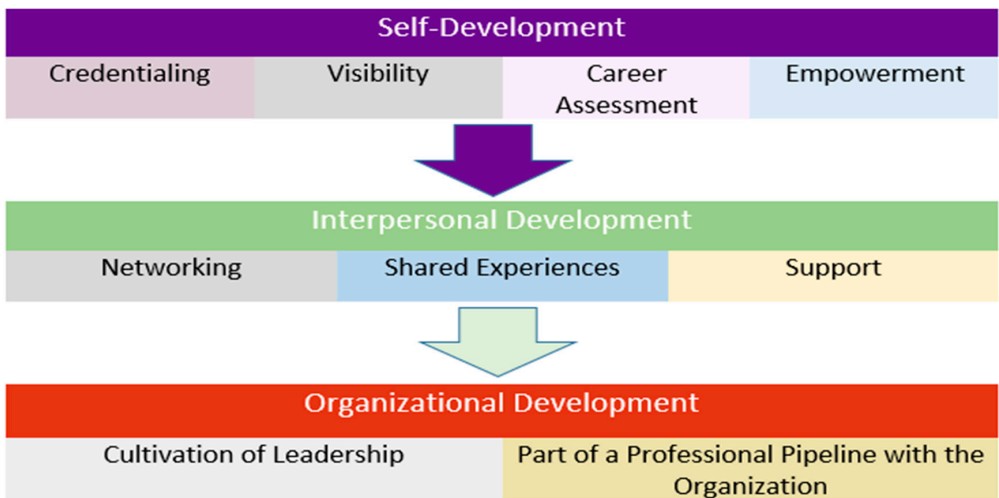

**Figure 2.** First Attempt at Defining Women's Leadership Levels and Themes.

In the next research iteration, the theoretical lens and methodology brought us to a far deeper understanding. Because we challenged ourselves to make sense of our experiences, and the data through a lens of Connected Knowing, we felt it could be an opportunity to be vulnerable with each other and ask hard questions of ourselves and each other. Common questions we would pose to ourselves in this process included: "Is that really it?" "Why do you say that?" "Help me understand." As a result, our learning evolved, moving from the initial finding of "Visibility" on the personal level, to "Being Seen and Heard." This reframing in Figure 1. elucidates the shift from the sole focus on the individual to an expectation placed on others and the organization to "recognize" and "actively support" us, thereby highlighting the need to attend to multi-level development. In addition, while Strengths quest was part of a WLP career assessment assignment, what we really needed was a structured space to reflect on our professional paths. This was reframed in Figure 1. as well. Lastly, we realized that we never felt truly empowered by our participation in WLP, and that theme was removed. This process included more generation of personal stories and accompanying introspection on identity to confront what we thought we knew about our experiences in the WLP program, and each other.

Figure 1 Themes at the Personal Development Level: Vulnerability, Structured Space for Reflection, and Being Seen and Heard.

Vulnerability involves allowing oneself to be wholly honest about one's experiences, open to emotional exposure, and risk of harm. Prevalent ideology sets up expectations that great leaders are strong and all-knowing—vulnerability is typically not understood as characteristic of a "good" leader. In Brene Brown's [41] book, *Daring Greatly*, she posits that women often feel they must hide important parts of themselves to mirror this expected type of leadership, but argues that vulnerability brings about creativity and change, allowing the leader to become stronger. In our experience, we all hoped to find a vulnerable space to explore work/life balance and integration, particularly around our motherhood identities. We needed to feel like we weren't alone in our personal experiences.

Having a structured space to reflect on our vulnerabilities as mothers allowed time that was critical, because finding time to reflect on personal and professional development is difficult to do in the context of our everyday lives. We were surprised that despite WLP being a women's-only program, structured opportunities to discuss the hot topic of work/life balance did not exist. To illustrate this point further, in our meaning-making process, we discussed "the secret." "Sometimes I would look at some of our senior administrative women, and they seemed to have it all . . . kids, spouses. I thought about the stress that I felt and internal conflicts I had with responsibilities of work and home. What are they doing that I am not doing? I wanted to learn the secret. They never seemed frazzled or stressed. They had great family stories. They were respected. It was as if they had the secret I didn't know. I had hoped that I too would learn the secret."

Another author noted that she hoped for rich dialogue about the socially constructed norms around work and caregiving. Such dialogue did not exist. Building intentional space for personal reflection is critical to fostering vulnerability to be seen as a strength, and when attempting to build relationships across differences. Bell et al. [42] came to a similar conclusion in their struggle interpreting research data as Black and White women sharing a reflective space. "Coming into the room as whole people—and demystifying what we imagined about one another both within and across race lines—helped move our conversations forward" [42]. Without this structured space for reflection built into the program's curriculum, our ability to share authentically across difference was thwarted. Our writing process opened up this opportunity for us as co-authors, but we wonder how we could have unpacked our differences—particularly around race—differently with more structured space for reflection in the WLP program.

Lastly, as previously noted, we merged the ideas of credentialing and visibility in Figure 2 to "being seen and heard" in Figure 1. This theme was central to our discussion on why we applied to the program. One of the articulated learning outcomes of the WLP program included maximizing influence with departmental stakeholders at the university. Thus, "I wanted to get on a senior woman leader's radar," said one author. Becoming visible as aspiring women leaders was critical to our personal development. We wanted to be seen and noticed as competent leaders with subject matter expertise and hoped this would lead to gaining sponsors. Having this affirmation from senior women leaders was crucial to our leadership efficacy, and being accepted into WLP offered a particular credentialing of sorts. Given the increased awareness regarding the importance of gaining a sponsor to put your name forward for opportunities, we saw the connection of being seen and heard to gaining more leadership opportunities. While we were acknowledged for participating in the program with a certificate, any correspondence shared with our supervisor did not produce the desired increase in being seen and heard.

Figure 1 Themes at the Interpersonal Level: Intersectionality, Supportive Community for Vulnerability and Networking.

The two most poignant examples that exemplified the need for discussing intersectionality were based on our racial and motherhood identities. We all hoped to find both space and the support to explore the work/life balance and integration, particularly around our motherhood identities. Haynes [43] used oral history to explore identities of women accountants, embodiment, and motherhood. She discussed gendered behavior, such as how to dress and the experience of pregnancy at work. Because this is such a strong part of women's socialization, we expected to discuss these things. During the processing of our stories, one author indicated a shocking revelation. She said, "I do not even put out pictures of my kids on my desk at work because I do not want people knowing I have four children. I am afraid people would perceive me as not having enough time to do the work." This reflection supports Gatrell's [44] work on maternal bodies in management studies, where she discusses the discounting of pregnant women's capabilities based on perceptions. We all worried that people would assume we had a "low work orientation" when we prioritized our children over work [44]. Throughout our research process, we had multiple conversations about "mom guilt" for choosing career over our children at times. Gatrell [44] also speaks quite a bit about breastfeeding

and disregard for the maternal body in managerial contexts. Because we were past this phase in our motherhood, we did not discuss this. However, it was noted that supportive supervisors who understood what it meant to be a working mom were critical to our overall experiences.

The message was clear to us: these struggles do not matter in organizational structures, and it is up to us as individuals to figure it out if we want to advance. While not all women are mothers or have family caregiving responsibilities, it can be a salient part of women's identities, and should at least be explored as relating to a woman's career path and in creating gender justice. It would have also been helpful to hear from women who did not have children by choice. Haynes [43] asserts "By failing to understand the impact of public and private, of professional and mothering identities, misunderstanding, reproduction of subtle forms of inequality, or even injustice may be perpetrated" [43] (p. 622). Given the prominence of this topic in popular culture and literature, it is a non-negotiable topic to include as part of the curriculum. Mothering and work identities are not separate, but entwined [43].

In terms of racial identities, the vast majority of participants in both cohorts were white women. Racial differences among the group were not acknowledged in the WLP program. However, this lack of acknowledgement does not erase the impact of race and racism on leadership development. The author who identifies as a woman of color shared an instance when a white male senior administrator commented on her hair. She said "While at commencement, I straightened my hair. When this man saw me, he said you should wear your hair like that all the time. It looks better. I felt like a lot of times I will not be accepted into higher leadership. I cannot be my authentic self. I have to burn my hair out to fit in." Her WLP cohort, with the support and structure of the WLP program, could have offered support and validation for her around this experience of racism. Additionally, as white women, we could have learned a lot about how our white privilege impacts our own leadership. Unfortunately, she didn't feel WLP offered her that safe space.

During WLP sessions, when discussing intersecting identities such as racial or ethnic identities, the conversation seemed to be redirected, as if race and racism were a distraction from the program's focus, and that these conversations were tangential. There was a particular instance in which a diverse panel of senior administrative women shared personal journeys and words of wisdom to WLP participants. Only one presenter was a woman of color; she was African American. "When she began to talk about her experiences around racism and how to deal with racism in our profession, the conversation was immediately moved to another conversation and she was cut-off. I found myself in the bathroom talking to another woman of color to discuss how we had faced this but there wasn't opportunity for us to hear how other people who had moved up had navigated this." For women of color, both racial and gender identities impact leadership development—omitting one necessarily negates the other. In fact, this holds true for all of us—we cannot explore our gendered experiences with leadership development, without simultaneously interrogating our multiple identities and how those inform and influence our leadership.

The need for a supportive community where vulnerability can be practiced is the second theme at the interpersonal development level. This theme is the catalyst for creating a culture of courage, where participants show up without the expectation of perfection. This is particularly important as strong leaders understand the role of vulnerability as connected to humility. Due to the lack of a community that was supportive of sharing vulnerabilities, one author felt "isolated", and her sense of leadership identity "more fragile", as a result of her participation. This impact translated to the interpersonal level, where relationships were kept at a superficial, surface level.

Vulnerability was not perceived as a strength in this women's-only leadership program. One author stated, "When doing an exercise on authentic leadership, I felt embarrassed to share that I was divorced and not married. I did not want to be perceived as less than perfect because I was so focused on trying to keep up with the other women leaders around me, who seemed to have it all together. But being divorced is a dominant identity among multiple identities in my lived experience." Building a strong community in which women could open up and be vulnerable

about personal challenges related to leadership could support personal self-efficacy, promote strong interpersonal relationships, and begin to highlight structural inequities—thus, connecting the personal to the political.

Networking, the third theme in interpersonal development, is essential to professional life. It leads to career advice, opening up professional opportunities, and learning from others about their leadership journey. More than countless hours at the computer, leadership is the ability to connect to others, incorporate outside perspectives, and navigate groups. We hoped that through participating in WLP, we would learn how to network strategically, given the program's articulated learning outcome to build critical networks and partnerships. It is important to note that women leaders may face unique challenges when it comes to networking. For instance, one author was "uncomfortable with the idea of networking", and preferred that it was framed as "relationship-building." This was consistent with Ely, Ibarra, and Kohl's [1] thoughts on women viewing networking as inauthentic. The issue of work-life conflicts was also applicable for us in relation to networking. "I have to pick up my kids from daycare right after work and do not have the freedom to attend a networking happy hour," said one author. As a group, we were often preoccupied with caretaking duties at home, and consistently struggled to devote networking time away from family or personal interests. Again, this topic was not addressed in the WLP program.

Figure 1 Themes at the Institutional Level: Cultivation of Authentic Leadership and Changed Culture—Diversified Pipeline within the Organization.

The two themes at the institutional level were discussed as aspirations for a strong women's leadership development program. The cultivation of authentic leadership requires that we bring more of who we are to the table more often. In order to be consistently authentic, we needed to explore and accept the complex identities of each WLP participant. This is why we acknowledge that the levels are interrelated. The cultivation of authentic leadership cannot exist without structured space for personal reflection on our multiple identities and supportive community for this. How can we be confident in being seen and heard within our university without being authentic? While there was knowledge gained from an expert speaker on Authentic Leadership, the didactic format of the content delivery made personal relevance difficult because there was very little time left for group processing and interaction. If authentic leadership is truly embraced at the organizational level, it should be fully integrated throughout the women's leadership development curriculum. Only then will cultures change and bring with it a diversified pipeline of women who have made efforts to understand our multiple, intersectional identities through the practice of vulnerable sharing in a supportive community. Considering the lack of women's leadership in senior positions at University X, the WLP program attempted to resolve the problem, but did not fully respect the diversity of the women in the pipeline. The institution must change the leadership culture to authentically value this diversity, or the pipeline will remain leaky.

The WLP program offered ripe opportunity to cultivate authentic leadership and changed culture and diversify our institutional talent pipeline. Had we, as WLP participants, been able to experience at the personal and interpersonal levels what has been discussed here as emergent themes—vulnerability, critical reflection, feelings of being seen and heard, exploration of our multiple identities and intersectionality, supportive community, and networking—a critical piece of this sort of authentic leadership and changed culture would have been seeded. After all, WLP participants represent an emergent leadership, and our cohort would have had the opportunity to truly learn alongside each other, gain insight from shared personal experiences around our multiple identities, and increase empathy related to navigating leadership. With this new understanding of the barriers and opportunities women face in the workplace—and in our institution, specifically—we would be in the unique position to inform and develop relevant policies and practices to support women's leadership development at the institutional level. We would be more empowered to do that, because of our WLP experience.

Based on our own discussion that emerged through our autoethnographical work, we imagine some of these institutional policies and practices would include: strengthened stop the tenure clock policies; more flexible work schedules for staff; more convenient and available lactation rooms; equal pay practices and policies; redesign of the review for tenure process which values gendered labor equally (i.e. service and student support); diverse and inclusive hiring; strengthened sexual and gender-based harassment policies and supports; increased resources allocated to women-only programs like WLP; and creating safe spaces for other affinity groups (including women of color, LGBT women, international women, women with disabilities). WLP participants would be wonderfully positioned to advocate for and lead such institutional changes and in doing so, create a more inclusive institutional climate.

## 8. Discussion

Our results supported Ely, Ibarra, and Kolb's [1] definition of women's leadership development, in that our experiences in the WLP program led us to see the critical role identity work plays in this process. This insight underscores the importance of the theoretical lens used in program design and delivery. The lens that informed our research analysis, Connected Knowing [30], was effective in our own positioning as "knowers", and facilitated a fruitful critique of our experiences and institutional context. In addition, pedagogical tools that align with that lens and facilitate deeper analysis are effective in the identity work of women's leadership development programs. For example, our use of autoethnography as an evaluative method was valuable in drawing connections between the personal and political, and the individual and structural. Therefore, program curricula should be far less concerned with content covered, and more focused on program approach and process. Carefully attending to ways in which power positions participants as knowers and leaders from the outset will significantly impact the participant's personal leadership development and, ultimately, positively impact the organization. While some individual skill-development, such as budgeting and negotiating, were helpful parts of the curriculum, we were seeking a more transformative approach that gets at the root of the problem of women's leadership development—gender inequity—and flips the power. The participants themselves become the knowers and experts. This insight informs our recommendations, and is further explored in the below section.

As such, there was certain content that was overlooked in WLP that we believe all women's leadership programs must address. Gender bias, as well as other forms of discrimination based on intersectional identities, is substantive in the literature and must be covered. Structured time for reflection should be spent on addressing the personal barriers women face in leadership development, such as finding a sponsor, and sharing these stories with others. This storytelling will build empathy and understanding, as well as highlight structural barriers in place that negatively impact gender equity. Additionally, networking opportunities with other aspiring and established women leaders will support efforts to advocate for policy changes that address the structural barriers to success. Women's-only programs uniquely promote these kinds of learning experiences. Women can bind together and use their power to support one another. By incorporating the sharing of personal experiences and listening to other stories of women into women's leadership development programs, we co-construct knowledge as Connected Knowers, and learn more than we could from an experience like this, had we been on our own. Through finding and trusting our own voices, we can develop a sense of leadership purpose and truly create institutional change.

We did not complete WLP with an earned sense of purpose as women's leaders, even though we were feeling the societal pressure to "lean in" due to the popularity of Sandberg's [2] book at the time. However, we did meet the program's goal of self-awareness in the end. Throughout our participation in the program, we identified with the struggle to have it all, and turned to the WLP program to seek mentoring and sponsorship. We needed the mentoring to answer some lingering questions regarding our multiple identities, such as managing work/life balance as mothers and navigating race and racism. We needed the network for sponsoring and the visibility of being seen

and heard as aspiring senior leaders, especially because we knew the climate was still chilly at the very top—and even more so for more marginalized, intersectional identities. In summary, our findings showed us that we all struggled with our own self-efficacy, because we did not have the structured space for vulnerable reflection that is needed to explore what we are capable of. It would have been especially helpful to interrogate this in a supportive community with other women leaders. We now know that we need other women to help realize our path to leadership success. Overall, our goal to make the program better for other women will come to fruition as the knowledge we have gained is applied. Next, we provide recommendations for future women's leadership development programs based on our analysis.

## 9. Recommendations

### 9.1. Recommendation 1: Practice Intersectionality in Women's-Only Leadership Development Programs

Based on our experiences in a women's-only leadership development program, we wonder about the value of a women's-only space in developing leaders. Being in our women's-only leadership development program was a critical part of our analysis. This issue is also debated in the leadership literature. Vinnicombe and Singh [26] discuss the importance of women's only leadership development programs. "Skeptics may argue that women-only programs do women a disservice", because it removes interactions with men that they will inevitably need to collaborate with [1]. A women-only space is an integral component to leadership development programs for women, as women value being seen as experts in their field, having an intrinsically interesting job, personal accomplishment, self-development, and balancing work and personal life. In contrast, men see career success in terms of climbing ladders and gaining influence, with the external trappings of success, including high salary, cars, and status [26].

Significantly, though, women-only programs should not be an add-women-and-stir approach, but should encourage women to explore how their gender, intersecting with other dimensions of their identity, influences their leadership for the quality improvement of programs [1]. For us, there was a disconnect between our hopes for what a women's only space would mean for us, and our actual experience of this gender-specific programming. For example, we expected that, participating in a women's development program, women would not be understood to be monolithic as a group. Interestingly, we learned that without complicating "women's" leadership development, we were reinforcing gender binaries. A subsequent risk of this sort of traditional practice is that women are essentialized and kept in our place. Identities are full of complexities. For example, even though all participants in the program were women-identified, some were classified as faculty and some were staff. One author was appalled at how this distinction was not engaged as a topic of conversation, especially as it relates to power dynamics. She said, "How could we sit there and pretend that there wasn't a real difference in how we were perceived in relation to women's leadership on campus, or even in that room? Why couldn't we discuss how to help each other within these clear power structures? How ironic that we were there to help women advance but didn't discuss the social inequity inherent in the program?"

We assumed a women's-only space would provide us much needed support for career and life reflection around our dominant and marginalized identities, promote authentic relationships, and empower us to create change as individuals, colleagues, and supervisors. This was not our experience. In fact, contrary to the ultimate goal of the program, in some ways we felt diminished in our growth and learning as a result of the program design. One author shared,

> "All of the sessions were completely didactic in their delivery. There was no acknowledgement that much of these uni-directional messages were coming from male guest speakers. This completely reinforced gender inequalities and dynamics related to women's leadership development."

Another commented, "Generally I'm resistant to traditional women's leadership programs because they reinforce institutional power structures, but I wanted to open myself up to the experience." Furthermore, diminished growth at the individual level extended to the interpersonal and organizational levels insofar as potential growth was stunted. Subsequently, the ability to influence or shape the organizational/institutional level was hampered. How could we create sustained, structural change that would lead to gender equity if we did not feel empowered or in authentic relationship to other women, even others who are arguably in the "pipeline" with us?

Women of color's experiences of the intersection of racism and sexism, along with other forms of identity oppression, were completely overlooked in our WLP experience. The message was clear to us: these struggles do not matter and it's up to us as individuals to figure it out if we want to advance. We recognized that which holds for white women does not necessarily hold for women of other racial backgrounds [45]. Given the data on how race impacts pay rates, tenure, and advancement, failing to attend to our racial and other identities was a significant missed opportunity to prepare us as effective change agents and leaders. As one author shared,

> "As a white woman who works actively to be an ally in racial justice work, I was so frustrated by the lack of engagement around intersecting identities, and how those intersections both inform social constructions of leadership and impact our own leadership practice. How can we be effective leaders in creating change if we cannot show up as our whole selves and see others as their whole selves, operating within complex systems? We've got to work towards a more nuanced understanding of leadership than simply 'add women and stir' if we're interested in creating transformative change. That's the only kind of change I'm interested in."

Nevertheless, through our research process, we have found value in having a women's-only space. The most significant determining factor in the identity-based programming is the theoretical framework and program design used. Merely recruiting all women to a program on leadership development risks enacting an additive/reductionist/diversity management approach to inclusion in which women's full, lived experiences are not central, and therefore women's ways of knowing and knowledge is not valued. Moreover, this approach further constrains and reinforces gender norms and therefore maintains the status quo, leaving established power structures unchallenged. To challenge gender norms and truly advance *all* women, it is critical to design a program that positions *all* women as active knowers, and engages them in the co-construction of shared knowledge with others. Women's leadership development can be significantly impacted when program conditions foster deep, transformational learning among the participants. As Ely et al. [1] argue, "Pedagogical theories, however, have failed to keep pace with practice. Practitioners and educators lack a coherent, theoretically-based and actionable framework for designing and delivering leadership programs for women" [1] (p. 6).

The question then becomes what conditions promote this sort of learning? We propose intersectionality as a theoretical framework and a key ingredient. It is worth defining what we mean by intersectionality, since this term is being increasingly used, arguably threatening its meaning and historical roots [46]. Intersectionality is defined as "the relationships among multiple dimensions of social relations" [47]. The concept of intersectionality was a major contribution from the fields of Critical Race Theory and Women's Studies, and arose out of a critique among women of color on theories of race and gender. Previous research solely focused on race or gender failed "to acknowledge lived experience at neglected points of intersection—ones that tended to reflect multiple subordinate locations as opposed to dominant locations." [47]. There is significant opportunity and need for educators to incorporate the idea of intersectionality in our scholarship, teaching, and campus work. Further, Debebe and Reinert [27] discuss how sociopolitical identities shape our actions as leaders. Consideration to the dynamic, interrelationships between and among *whole* individuals and work environments is critical in leadership development programming.

Translating a theoretical framework and concept into practice through programmatic design and implementation can be challenging. One key, practical step is to engage stakeholders in all stages of the program including development, implementation, and evaluation to ensure the program speaks to participant needs as developing leaders. Deliberate attention to multiple, intersecting identities should be evidenced in all aspects of program development, implementation, and evaluation in terms of both curriculum and pedagogy. A suggested, guiding question to help focus this attention is, "To which women is this program speaking? Who is being excluded?" For example, what is meant by "woman?" Does that include only cisgender women and/or trans women as well? If so, a first step would be to include gender inclusive language in application materials, such as the options to choose preferred gender pronouns. In sum, the longstanding intersectional approach held by feminist thinkers provides a much deeper understanding of oppressed identities, and should be used to frame women's leadership development programming, in order to result in most significant impact.

It is important to note that we did not arrive at this recommendation easily. Our selected methodology truly facilitated a challenging, critical, introspective, and enlightening process. As a result, we came to appreciate autoethnography both as fruitful research process, and also a key finding and recommendation.

### 9.2. Recommendation 2: Autoethnography as Methodology for Program Evaluation

Autoethnography is described in the *Handbook of Autoethnography* this way:

"Autoethnography is not simply a way of knowing about the world; it has become a way of being in the world, one that requires living consciously, emotionally, reflexively. It asks that we not only examine our lives but also consider how and why we think, act, and feel as we do. Autoethnography requires that we observe ourselves observing, that we interrogate what we think and believe and that we challenge our own assumptions, asking over and over if we have penetrated as many layers of our own defenses, fears, and insecurities as our project requires. It asks that we rethink and revise our lives, making conscious decision about who and how we want to be. And in the process, it seeks a story that is hopeful, where authors ultimately write themselves as survivors of the story they are living" [48].

Autoethnography promotes introspective self-reflexivity, in which personal narratives are considered and reconstructed to create meaning. Therefore, we argue that autoethnography holds great potential as an integrated methodological tool, in both program evaluation and in the development of women leaders through fostering transformative learning. It is important to note that the same could be said for CAE.

Program evaluation is crafted around quantitative outcomes necessary to meet the inquiries of stakeholders. Norris and Kushner [49] note that evaluation is often politicized as a tool of program funders and is viewed through a narrow, reductionist lens of accountability, used only to demonstrate 'evidence' of impact, thus missing its potential to contribute to learning [50]. Traditional measurements of program success measured by productivity or efficiency are often byproducts of capitalistic patriarchal culture. Chapman [51] asserts that ignoring unintended consequences and measuring only outcomes in program evaluation blocks learning. If the evaluation does not seek reflective, authentic feedback, responses fail to get at the underlying assumptions that frame program design, implementation. As a result, such evaluation practices maintain status quo, reinforcing these assumptions. We live in a male-dominated world where receiving reward is prioritized; and we are rewarded by saying and doing what people expect.

In addition, the design and delivery of the program evaluation itself determines both the impact of the program and the assessment findings. For example, offering limited space and time to provide feedback on program evaluation only allows for quick, superficial thinking, without appropriate time for depth of reflection. Conventional ways of thinking and conducting program evaluation are often narrow and limiting, both in terms of design and the findings subsequently yielded. By intentionally

providing time and space for participants to describe and analyze personal narratives, both the process and a product of evaluating the leadership development experience are realized. For instance, Hawthorne Calizo [24] conducted an unpublished dissertation case study on the WLP program, finding that supervisors were supportive of participation, but not engaged in the participant's learning—the missed opportunity to share program evaluation findings and participant learning with stakeholders directly impacts opportunities for organizational growth.

By contrast, Burns [52] asserts that "we need to look wider than causal attribution, beyond numbers and beyond traditional qualitative material to understand the dynamics of a process, not to ask what's happening, but how and why it is happening" [52] (p. 6). Jarvis et al. [29] discuss the application of relational approaches to leadership development program evaluation, and suggest using a Complex Responsive Process theory (CRP) as a lens applied to evaluation. This suggests an approach based on relationships and trust as essential to sharing authentic accounts of experience and uncovering collective wisdom, underpinned by an approach to leadership development that values different domains of expertise and the importance of 'connecting' and peer-to-peer spaces. This resonates with our recommendation to use autoethnography.

Custer [53] found seven benefits of applying autoethnography to professional situations: 1. it changes time, 2. requires vulnerability, 3. fosters empathy, 4. embodies creativity, 5. eliminates boundaries, 6. invites and honors subjectivity, and 7. provides therapeutic benefits .We were able to utilize all of these elements to reflect on our experience from different angles and struggled with our vulnerability in particular, as we had yet to obtain a deeper understanding of our own and other other's personal story.

As we listened to each person's response to the guiding questions, put ourselves into each other's frame of reference, and allied ourselves to each other's views with empathy (not judgment), we discovered our own truths about our leadership identities. Our learning and understanding deepened, as we did not have to compromise our beliefs or leadership styles to participate. Our vulnerability allowed us to clarify our leadership purpose. Autoethnography allowed us to revisit the past, and make meaning of the impact of our experience in WLP on ourselves and our institution. Therefore, we recommend the use of autoethnography as a program strategy to develop women leaders and support relationship building. Furthermore, because it promotes authentic relationship building, connections between the personal and political are highlighted. This increased awareness and deep understanding of larger, systemic barriers and opportunities will more significantly impact organizational and institutional development, through participants' own leadership practice.

In summary, conducting reflective research (like autoethnography) is outside of the scope of traditional quantitative methodologies, and offers the inclusion of all women's voices and identities that are critical to effective implementation and evaluation of women's leadership development programs. It is worth noting, however, that the method of autoethnography has limitations. For example, it requires that participants be open and truthful about personal experiences. There are always power dynamics at play that can create resistance to this process. While we believe that this recommendation makes a methodological contribution to the literature on women's leadership development, we by no means see it as the panacea, and recognize the potential for negative outcomes, as explicated earlier when addressing ethical considerations.

## 10. Conclusions

As our research process reminded us, authentic leadership requires relationship building and trust in others. Employing autoethnography as a methodological tool and connected knowing as our theoretical lens allowed us the opportunity to be vulnerable with one another and share our personal stories about our participant experiences in WLP, a women's only leadership development program at University X. As a result, we moved from an initial, surface level exploration of our WLP experiences to a much deeper understanding of *how* and *why* our experiences played out as they did, granting us deeper insight and transformative learning. It is critical to note that our learning is the result of



our intentional work to forge this shared reflection space ourselves, outside the parameters of the program, and in spite of the program design and previous methods of evaluation. Therefore, those who oversee women's leadership development programs should prioritize relationship building as a key strategy. The investment we made in relationship building and research methodology was fruitful, revealing a call for a multi-level approach to women's leadership development, including the personal, interpersonal, and organizational contexts. These levels are interconnected so that outcomes for each are dependent on the other, and determined by the effectiveness of relationship building. If the advancement of all women is a goal at the organizational level, practicing intersectionality from start to finish in women's leadership development programs, and including autoethnography as an evaluation method, are two suggested strategies that can effectively stop the leaky pipeline. In a program that includes these recommendations, women would be empowered to be their whole selves without restriction; there would be more visible and tangible mentoring and sponsoring of other women. We would know that women's leadership would be improved because we would see social, institutional, and cultural factors change. For example, there would be a better understanding of women leader's identities as mothers, and as having differing dominant and subordinate identities in general. Evidence of this expanded understanding, and appreciation of women's leadership, would be in the membership of the program itself with participants from different positions, and with diverse social identities. No longer would there be an artificial separation between the personal and the political. *All* women would have agency as leaders. Their voices would be lifted from the trenches.

**Author Contributions:** Robin Selzer designed the research study and identified the theoretical lens. All authors contributed equally to conducting the research, analyzing the data, and writing the paper. All authors read and approved the final manuscript.

**Conflicts of Interest:** The authors declare no conflict of interest.

## Appendix A

Appendix for American University Terms:

Adjunct instructors are hired by colleges and universities on a contractual, part-time basis.
Tenure can be defined as a permanent professorial appointment.
Provosts are senior administrative officers in a college or university.
Faculty perform teaching, research, and service functions and staff perform administrative and support functions.

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
