# Peer review of "Rethinking Women’s Leadership Development: Voices from the Trenches"

_admsci, doi:10.3390/admsci7020018_

Round 1

Reviewer 1 Report

I am a women's leadership scholar and know the literature well. I was excited to see the title and get some real insight about a women's leadership program and evaluation. However, my experience reviewing was mixed. Here are some details:

1) There are small editing issues throughout (the piece needs a good edit before resubmission)

2) I really appreciated pages 1-7; the introduction, methods, methodology were outlined very well. Even though a small sample (3 people) you "case" on why it would be so powerful was very compelling. I know this methodology and believe it can be very effective. I was really excited to get to the findings once everything was laid out and the stage was set.

3) Then I got to the themes and charts, and in my opinion that is where things changed. I actually had to read that part 3 times to figure out what was happening. The charts were not well done and just had lots of words, but they were never used. You set the framework for all of the results of the study--and then what? I would have expected the rest of the findings to actually carefully walk through the figures and show examples of each one of the elements. Giving both charts up front just doesn't work. There were so many things outlined in the graphs that are never discussed at all.

4) I really don't see alignment between the findings (themes) and the discussion/recommendations. This all needs to be carefully crafted for it to be useful. Don't use the charts if you are not going to interrogate what all of the pieces mean. If they relate to things in your literature review, you must have them very clearly aligned and related. The charts didn't seem to guide the structure of your discussion either. The recommendations should be in its own section at the end. Intersectionality comes out of no where. You really cannot give recommendations unless readers get exactly who they emerged from the results of the research. A few kind of do--but not in a tight, organized way.

5) You missed the core of the article in my opinion. You built up how wonderful and important the methodology was and how important the "space" was for the three of you--but then we didn't really get the results. The charts just don't emerge from what your words say and what you outlined it to be about. I would have expected (at page 7) for a tight outline of the emerging theme categories and then a walk through of the results--any chart should reflect that. The results need to be much more in-depth with sections so each piece relates to your goal of the study. Some of the discussion is off topic--it doesn't tightly relate to the program. Everything should be about the program and your research questions. You need to have most of the data that emerged in a findings section without the literature--that is discussion. You need a lot more article space to actually give the results.

6) Your recommendations are not credible to me as I didn't see how they really emerged. The themes and discussion were really a let-down from what you built the paper to be. Help the readers understand the rich data that emerged from your experiences with many more details. Your paper needs to tell a story from the start to the finish. You were great until page 7. Your results need to be profound and we should really understand a lot of depth. You don't have time for the section on autoethnography. You need so much more in the findings section. Again (charts) don't provide charts with elements that you aren't going to full interrogate.

I do think this could be an important piece if it is revised heavily so that it is better aligned and there is much more depth. I encourage you to revise.

Author Response

The co-authors split up the tasks for suggested revision. A major focus was on writing a Results section. We used chart 2 to guide the framework for this section by writing an introductory paragraph to outline the levels (personal development, interpersonal development, and organizational development) and accompanying themes in each level one-by-one, using examples and our own words/quotes (many of these were already included in the Recommendations section-so we could see how the Reviewers thought this was convoluted).

In terms of the charts, we changed the word “self-development” to “personal development” at the first level and added double arrows to the chart to show that the levels are inter-related and inter-dependent. The spelling correction pointed out by Reviewer 2 was made as well.

Since pages 1-7 (intro, theoretical lens, and methodology) were effective, the only alteration we made was on page 5. Global women’s leadership literature (Fitzgerald, Morley, O’Connor) was reviewed and incorporated per Reviewer 2’s suggestion. We clarified that this study was located in several places including the abstract.

Clarification was provided about ethical considerations for the study, including why author’s identities’ are not provided with their quotes. We also clarified that the methodology was understood as feminist on page 17.

The Discussion/Recommendations section was separated into two sections. The Discussion section was written to connect what we found back to the literature and provide a smooth transition into the Recommendations section. The Recommendations section was reviewed to ensure that each recommendation emerged from the results section in a clear way.

Finally, a significant amount of time was spent cleaning up the References section, which included correcting dates, authors names, correcting APA format, and ensuring each in text citation was in the Reference list and vice versa.

Reviewer 2 Report

This is a valuable study of three women's experience of a leadership development programme in their own institution. The voices of the women are strong as would be expected in an autoethnoography. They are also critical of the programme with respect to a lack of opportunities for dialogic and reflective engagement, the pedagogical approach, non-recognition of intersectionality, and non-recognition of women's issues i.e. balancing family and career. However, there remain a number of matters that require attention before this work could be published. I am hopeful the comments will help the authors to refine their work. The article could be strengthened by: 1) locating the study in the literature about women's leadership in Higher Education such as in Australia by Jill Blackmore, Tanya Fitzgerald, Kate White; in the UK by Louise Morley; in Ireland by Pat O'Connor. There is a special issue of Gender and Education that focuses on HE that might prove useful. Gender, Work and Organisation is another potential source of literature. 2) considering the implications of what is revealed in autoethnographical research with respect to career futures 3) providing detail about ethical approval for the study to take place 4) clarifying which of the authors' words are quoted (this could be problematic as the authors' identities will be made clear on publication) 5) locating the study in the US - I have assumed that 6) considering the alternative metaphors to 'glass ceiling' such as 'labyrinth' (Eagly and Carli's work is cited 7) proof reading for typos (spelling of visibility in the chart), missed page numbers for direct quotations, order of authors, references not listed but cited (Belenky 1986) and vice versa (Silva and Carter), distinguishing between 2 works by Haynes (2008) by using letters, ordering of references alphabetically 8) shortening the very long quotations 9) identifying earlier that this focuses on 'staff' not academic faculty 10) recognising the essentialism in some of the literature cited (particularly as it is tentatively problematised by the authors) 11) locating the chosen methodology as 'feminist' 12) clarifying the marital status of the authors i.e. identified as married then divorced Many of these things are presentational matters and are easily rectified. Other things such as cited existing work in the field are more substantial. Hence my recommendation that major revisions are required.

Author Response

(The authors gave the same response as above.)

Round 2

Reviewer 2 Report

I very much enjoyed reviewing this article. My concerns about ethics remain with respect to revealing the University by virtue of your authorship as employees. There may be a possible impact on your professional reputations in the workplace. 

Author Response

Changed suggested revisions and addressed ethical concern by adding the following statement:

 "We understand that once the study is published we will lose this anonymity in some ways and also connect ourselves to the University of X. Yet, these stories are important to tell to help women and women’s leadership development programs."